# The Epidemiology, Management and Therapeutic Outcomes of Subdural Empyema in Neonates with Acute Bacterial Meningitis

**DOI:** 10.3390/antibiotics13040377

**Published:** 2024-04-21

**Authors:** Wei-Ju Lee, Ming-Horng Tsai, Jen-Fu Hsu, Shih-Ming Chu, Chih-Chen Chen, Peng-Hong Yang, Hsuan-Rong Huang, Miao-Ching Chi, Chiang-Wen Lee, Mei-Chen Ou-Yang

**Affiliations:** 1Division of Pediatric Emergency Medicine, Department of Pediatrics, Chang Gung Memorial Hospital, Chiayi 613, Taiwan; weiju21@cgmh.org.tw; 2College of Medicine, Chang Gung University, Taoyuan 333, Taiwan; mingmin.tw@yahoo.com.tw (M.-H.T.); jeff0724@cgmh.org.tw (J.-F.H.); kz6579@cgmh.org.tw (S.-M.C.); charllysc@cgmh.org.tw (C.-C.C.); ph6619@cgmh.org.tw (P.-H.Y.); qbonbon@gmail.com (H.-R.H.); 3Division of Neonatology and Pediatric Hematology/Oncology, Department of Pediatrics, Chang Gung Memorial Hospital, Yunlin 638, Taiwan; 4Division of Pediatric Neonatology, Department of Pediatrics, Chang Gung Memorial Hospital, Kaohsiung 833, Taiwan; 5Department of Respiratory Care, Chang Gung University of Science and Technology, Chiayi 61363, Taiwan; mcc@cgmh.org.tw (M.-C.C.); cwlee@cgust.edu.tw (C.-W.L.); 6Chronic Diseases and Health Promotion Research Center, Chang Gung University of Science and Technology, Chiayi 61363, Taiwan

**Keywords:** subdural empyema, bacterial meningitis, neurological sequelae, late-onset sepsis, *Streptococcus agalactiae*

## Abstract

**Background:** Subdural empyema is one of the more serious complications of bacterial meningitis and therapeutic challenges to clinicians. We aimed to evaluate the clinical characteristics, treatment, and outcome of subdural empyema in neonates with bacterial meningitis. **Methods:** A retrospective cohort study was conducted in two medical centers in Taiwan that enrolled all cases of neonates with subdural empyema after bacterial meningitis between 2003 and 2020. **Results:** Subdural empyema was diagnosed in 27 of 153 (17.6%) neonates with acute bacterial meningitis compared with cases of meningitis without subdural empyema. The demographics and pathogen distributions were comparable between the study group and the controls, but neonates with subdural empyema were significantly more likely to have clinical manifestations of fever (85.2%) and seizure (81.5%) (both *p* values < 0.05). The cerebrospinal fluid results of neonates with subdural empyema showed significantly higher white blood cell counts, lower glucose levels and higher protein levels (*p* = 0.011, 0.003 and 0.006, respectively). Neonates with subdural empyema had a significantly higher rate of neurological complications, especially subdural effusions and periventricular leukomalacia. Although the final mortality rate was not increased in neonates with subdural empyema when compared with the controls, they were often treated much longer and had a high rate of long-term neurological sequelae. **Conclusions:** Subdural empyema is not uncommon in neonates with acute bacterial meningitis and was associated with a high risk of neurological complications, although it does not significantly increase the final mortality rate. Close monitoring of the occurrence of subdural empyema is required, and appropriate long-term antibiotic treatment after surgical intervention may lead to optimized outcomes.

## 1. Introduction

Bacterial meningitis in neonates is associated with high mortality and morbidity rates [1,2]. *Streptococcus agalactiae* (Group B *Streptococcus*, GBS) and *Escherichia coli* are the most common pathogens that cause 65–78% of all neonatal meningitis, with high mortality rates ranging from 8–13% and 14–25% in term-born and preterm infants, respectively [2,3,4,5,6]. Bacterial meningitis is associated with a high risk of neurological complications, and neonates with meningitis are at high risk of long-term neurological sequelae [5,6,7,8]. Recent reports have found that 42.5–54% of survivors who experience bacterial meningitis during their neonatal period will have various degrees of neurodevelopmental delay in preschool age [9,10].

Most studies regarding neonatal meningitis focus on the microbiology, epidemiology, clinical manifestations and outcomes of GBS diseases, and fewer studies are for *E. coli* meningitis [3,4,5,6,7,8,11,12,13]. Very few studies have examined the neurological complications of acute bacterial meningitis in neonates [11,12,13]. Subdural empyema, one of the important complications of bacterial meningitis, is characterized by a suppurative collection between the dura mater and the arachnoid mater and a prolonged course of treatment and hospitalization [14,15]. Previous studies have concluded that subdural empyema is a neurosurgical emergency, potentially fatal, requiring a high need for clinical awareness. [14,15,16]. A recent surveillance study found that subdural effusion and/or empyema was noted in 21.7% of pediatric patients with acute bacterial meningitis, accounting for the most common intracranial complications [17]. However, the presence of subdural empyema may be underdiagnosed, and few studies have focused on this issue [18,19,20,21]. In this study, we aimed to describe the clinical features, microbiology, therapeutic strategies and outcomes of neonatal meningitis complicated with subdural empyema from two tertiary level medical centers in Taiwan.

## 2. Methods

### 2.1. Patients, Study Design and Settings

Between January 2003 and December 2020, all neonates aged less than 90 days old with documented bacterial meningitis were retrieved from the neonatal intensive care unit (NICU) database of Chang Gung Memorial Hospital (CGMH) to analyze the presence of subdural empyema after meningitis. Both the Linkou and Kaohsiung NICUs are tertiary-level medical centers located in the north and south of Taiwan, respectively. There are a total of 80 beds equipped with ventilators and 90 beds of special care nurseries. The overall annual admission in the NICUs of Linkou and Kaohsiung CGMH was more than 1000 critically ill and preterm neonates. Neonates with acute bacterial meningitis without subdural empyema were also analyzed as the controls for comparisons. Only cases of primary bacterial meningitis and subdural empyema after primary meningitis were enrolled for analyses. Neonates with central nervous system (CNS) infections after artificial devices, including ventriculoperitoneal shunt, extraventricular drainage and post-neurosurgical CNS infections, were excluded from this study. Cases of subdural empyema after sinusitis, neurosurgery, head injury or other disease entities were not included in this study. The electronic records of patients’ demographics, clinical manifestations, hospital courses, treatments and outcomes were all reviewed and recorded. This study was approved by the Institutional Review Board of CGMH (certificate no. 202201668B0), and a waiver of informed consent for anonymous data collection was approved.

### 2.2. Definitions and Data Collection

The presence of subdural empyema was confirmed after a radiologist had examined the neuroimaging studies, which showed a crescent or ellipse-shaped fluid collection in the subdural space. All subdural fluid collections were considered empyema because the sterile subdural fluid collection is mostly noted in chronic disease and rare in neonates. Meningitis was defined based on the criteria of the World Health Organization as the presence of positive cerebrospinal fluid (CSF) cultures for bacterial pathogens plus clinical signs and symptoms compatible with central nervous system infections [22,23]. In cases with negative CSF culture for bacterial pathogens, the following were considered bacterial meningitis: if the presence of clinical manifestations was compatible with bacterial meningitis plus a positive blood culture and CSF results showing at least one individual marker of bacterial meningitis, including a leukocyte count of more than 2000/μL, a glucose level of less than 34 mg/dL [1.9 mol/L], a ratio of CSF glucose to blood glucose of less than 0.23, or a protein level of more than 220 mg/dL [22]. Severe sepsis, septic shock, and disseminated intravascular coagulopathy were defined based on the criteria of the Centers for Disease Control and Prevention [22].

The presence of neurological complications and long-term neurological sequelae in these patients were evaluated based on the definitions of our previous studies [21,24]. In our institute, transcranial sonography was routinely performed for all patients with bacterial meningitis, and brain computed tomography (CT) scans or magnetic resonance imaging (MRI) were arranged after the attending physicians’ clinical evaluation, especially when clinical consciousness changed, seizures, abnormal neurological symptoms, or abnormal sonography were noted. The follow-up neuroimaging studies for neonates with bacterial meningitis, including transcranial ultrasound, brain CT scan or MRI, were also decided by the attending physicians, who always closely observed any neurological symptoms or signs or when intracranial abnormalities were suspected. The neurological complications after meningitis and after the onset of subdural empyema were also recorded. The onset of subdural empyema was defined as the day when the first neuroimaging studies identified abnormalities.

All patients’ demographics, clinical manifestations at the onset of meningitis, laboratory findings at the onset of symptoms and treatment course, neurological findings at discharge and final outcomes were recorded. In our institute, survivors of neonatal meningitis were followed by a neurologist after discharge for standard neurological examinations, and serial outcomes were graded according to the validated Pediatric Version of the Glasgow Outcome Scale (GOS-E Peds) [25]. A favorable outcome was defined as a GOS-E Peds score of 5 (good recovery) and an unfavorable outcome as a score of 1 (indicating death) to 4 (moderate disability). In this study, all mortality cases and neonates with critical discharge on request and those with early signs of cerebral palsy at discharge were considered unfavorable outcomes.

### 2.3. Antimicrobial Susceptibility Testing

All the bacterial pathogens that caused neonatal meningitis were tested to check the presence of antibiotic resistance. Antimicrobial susceptibility testing was performed with the disk diffusion method, as described in previous studies [26]. For gram-negative bacteria, antibiotic susceptibility patterns were determined according to methods recommended by the National Committee for Clinical Laboratory Standards Institute (CLSI) for disk diffusion methods and categorical assignment was carried out using CLSI breakpoints [27]. All the following antibiotic susceptibilities were tested against GBS isolates: ampicillin, macrolide, clindamycin, vancomycin, penicillin, cefotaxime and teicoplanin. The guidelines of the CLSI for the disk dilution method were used [27].

### 2.4. Statistical Analysis

Neonates with subdural empyema after acute bacterial meningitis were in the study group, and those without subdural empyema were also enrolled as controls. The clinical features, microbiology, treatments and outcomes were compared to characterize the cases of subdural empyema in this study. Categorical and continuous variables were expressed as proportions and medians (interquartile, IQR), respectively. The χ^2^ test or Fisher’s exact test was used to compare the categorical variables, with odds ratios (ORs) and 95% confidence intervals (CIs) calculated in the results. The Mann—Whitney *U*-test and the *t*-test were used to investigate the differences between continuous variables depending on the distributions of the data. *p* values of <0.05 were considered statistically significant.

We aimed to investigate the independent risk factor for occurrence of subdural empyema in neonates with acute bacterial meningitis. Associations between patients’ demographic, clinical and molecular characteristics, therapeutic strategies and laboratory results were tested in univariate analyses, and odds ratios (ORs) were used to quantify the strength of associations. Covariates presumed to be associated with final adverse outcomes based on previous studies [3,12,21] and those associated with mortality at *p* < 0.1 were subsequently entered into the multivariable logistic regression model. All statistical analyses were analyzed using SPSS version 21.0 (IBM SPSS Statistics, Chicago, IL, USA).

## 3. Results

### 3.1. Patients’ Demographics, Microbiology, and Clinical Features

During the study period, a total of 153 neonates with bacterial meningitis were identified from the NICUs of the two medical centers in Taiwan. In the cohort, a total of 27 neonates (17.6%) had subdural empyema after acute bacterial meningitis. Neonates with subdural empyema after bacterial meningitis had comparable demographics with those without subdural empyema after bacterial meningitis (Table 1). The median gestational age and birth body weight in the cases were 37.3 ± 2.09 weeks and 2826 ± 535.9 g, respectively. Only a few cases of neonates (29.6%, *n* = 8) were preterm (GA < 37 weeks), and only 22.2% (*n* = 6) cases were low birth weight infants (BBW < 2500 g). The median age at onset and diagnosis of bacterial meningitis and subdural empyema was 30 (21.0–47.0) [Interquartile range (IQR)] and 38 (25.0–60.0) days old, respectively. The average occurrence of subdural empyema from the onset of acute meningitis was 9.6 ± 4.7 days. There were 3 (11.1%) cases of early-onset disease (EOS, onset at age ≤ 7 days old) and 24 (88.9%) cases occurred between 7–90 days old (late-onset disease, [LOS]). Most patients were inborn, and only 11.1% (*n* = 3) were transferred from other hospitals.

In the overall cohort of neonates with meningitis, 92.8% (*n* = 142) patients had a positive CSF culture and 128 (83.7%) neonates had concurrent bacteremia (positive blood culture). All neonates with subdural empyema had positive CSF cultures. The pathogen distributions of cases with subdural empyema were similar to that of neonatal meningitis. A total of 44.4% (*n* = 12) of cases were caused by group B *Streptococcus* (GBS), and 33.3% (*n* = 9) were caused by *E coli*. Other pathogens included *Klebsiella* spp. (7.4%, *n* = 2), *Staphylococcus aureus* (7.4%, *n* = 2), *Salmonella* (3.7%, *n* = 1), and *Pseudomonas* spp. (3.7%, *n* = 1). Most neonates (88.9%, *n* = 24) with subdural empyema had concurrent bacteremia at the onset of bacterial meningitis, and all of the blood cultures were the same as those identified in the CSF culture. All neonates with meningitis had a cranial sonography examination following bacterial meningitis, and cranial imaging studies were performed in 98 (64.1%) of 153 patients, and the median (IQR) days between the lumbar puncture and first neuroimaging was 12.5 (7.0–18.0) days.

At the onset of subdural empyema, which was defined when the first brain imaging studies were found to have the typical crescent- or ellipse-shaped fluid collections (Figure 1), most patients had a fever (85.2%), seizure (81.5%), feeding difficulties (66.7%) and respiratory symptoms (63.0%). Most clinical manifestations were not significantly different between neonates with subdural empyema and the controls. The laboratory results were also comparable between neonates with subdural empyema and those with meningitis but no subdural empyema. As to the CSF examinations, neonates with subdural empyema had a significantly higher white blood cell count, a lower glucose value, and a higher protein value than the controls (*p* = 0.014, 0.004 and 0.015, respectively) (Table 1).

### 3.2. Therapeutic Courses and Neurological Complications

In neonates with subdural empyema after bacterial meningitis, the common manifestations were persistent symptoms despite appropriate antibiotic treatment (*n* = 27, 100%). Neonates had persistent fever, seizure, and sometimes clinical deteriorations, including septic shock, respiratory failure and decreased level of consciousness. However, none of the cases had transtentorial cerebral herniation, pupil dilation or abnormal posturing during the hospitalization. In our institute, ampicillin plus cefotaxime or vancomycin plus cefotaxime were usually prescribed as empiric antibiotics, and modification of therapeutic antibiotics depended on the CSF culture results and the attending physicians’ decisions. Because only 11.1% (*n* = 3) of neonates with subdural empyema had antibiotic-resistant pathogens that required modification of initially prescribed antibiotics, none of the cohort cases had delayed treatment with adequate antibiotics. During the therapeutic courses, neonates with subdural empyema were significantly more likely to have neurological complications than those without subdural empyema, especially seizure, subdural effusion and periventricular leukomalacia (Table 2).

### 3.3. Therapeutic Strategies and Final Outcomes

The time course of all neonates with subdural empyema is summarized in Figure 2. The duration of antibiotic use was significantly longer in neonates with subdural empyema than in cases of bacterial meningitis but no subdural empyema [median (IQR) 27.0 (24.0–65.0) days vs. 22.0 (19.5–55.0) days, *p* < 0.001]. There were two patients (7.4%) who died, and the mortality rate was not significantly higher. Surgical interventions were arranged in all neonates with subdural empyema. At discharge, 56.0% (14 out of 25) of the survivors had neurological sequelae, which was comparable with those without subdural empyema.

We aimed to investigate whether the presence of subdural empyema was independently associated with final adverse outcomes, which included those with in-hospital mortality and neurological sequelae at discharge (Table 3). Preterm neonates (GA < 37 weeks) had a significantly higher risk of worse outcomes than term-born neonates. Variables characterizing neonates with early-onset disease, seizure at onset (defined as seizure attack within 48 h after onset of meningitis), respiratory failure and septic shock, and presence of subdural empyema that required surgical intervention were entered into the multivariable regression model. After multivariate logistic regression analyses, the independent risk factors for final adverse outcomes in neonates with acute bacterial meningitis were seizure at onset (OR, 2.45; 95% CI: 1.15–4.56, *p* = 0.013) and early-onset sepsis (OR, 3.38; 95% CI: 1.10–8.04, *p* = 0.027).

## 4. Discussion

To our knowledge, this is the first study to characterize subdural empyema in neonates after acute bacterial meningitis, although some case series can be found in the literature [15,18,19]. This study shows that subdural empyema complicates 18% of all neonates with bacterial meningitis. It has been well known that most neonates with bacterial meningitis have neurological complications, and nearly half of the survivors have long-term neurological sequela. However, we found that neonates with subdural empyema had comparable final outcomes when compared with the controls, similar to those reported in the literature [14,20,28,29,30]. The diagnosis of subdural empyema was often delayed when neonates were on antibiotic treatment, and some of the neurological complications were noted even several weeks after the onset of neonatal meningitis. Therefore, continuous monitoring and high alertness of relevant symptoms and signs are important for clinicians.

The pathogen distributions of neonates with subdural empyema after bacterial meningitis in the cohort were similar to those of neonatal meningitis [4,5,6,8,12], and none of our cases had multiple microorganisms isolated. Previous studies found *Streptococcus agalactiae*, *Listeria monocytogenes* and Enterobacteriaceae to be important pathogens of subdural empyema in neonates with bacterial meningitis [29,31,32], although some important pathogens, including *Haemophilus influenza* type b, *S. aureus*, *Salmonella* spp. and *Neisseria meningitis* have been reported in the literature [15,18,19,33]. The incidence of neonatal GBS sepsis has declined in the past decade, mostly due to current infection control and preventive strategies that have worked successfully to reduce maternal colonization and vertical transmission [7,34]. We found that Gram-negative bacilli have become important pathogens of neonatal meningitis [21,31]. Most of the Gram-negative bacteria were antibiotic-susceptible, and initial inadequate antibiotic treatment was rare in our cases at the onset of meningitis. Therefore, we do not conclude that the occurrence of subdural empyema is related to initial delayed antibiotic treatment and suggest it occurs spontaneously. Additionally, we also found that timely surgical decompression was the most important factor; initial antibiotic management was not associated with final outcomes, which was similar to previous studies [14,20,29].

The clinical features of our cases with subdural empyema were compatible with those reported in the literature [20,23,35]. A recent study concluded that subdural empyema should be considered in patients with persistent fever while on adequate antibiotic therapy or above 4.65 protein-to-glucose ratio in the CSF [16]. Among all meningitis-related complications, subdural empyema was often identified earlier [21]. In adult patients, subdural empyema should be suspected when clinical deterioration is noted within the first 8 h after lumbar puncture [36]. However, local expansion of the empyema that caused brain shift or transtentorial cerebral herniation was rarely observed in our cases. Only one of the cases of subdural empyema after severe subdural effusion had mild brain shift (Figure 3). Therefore, the initial presentations can be subtle. A decision must be made for highly suspicious presentations to arrange intracranial images or continuous monitoring, which is necessary for prompt intervention after early diagnosis [23,24].

In our cohort, cases of subdural empyema had a higher WBC count, lower glucose level and higher protein level, which indicated more severe intracranial inflammation [6,16,37]. Most of the subdural empyema were direct bacterial invasions, and it is reasonable that these patients were more likely to have concurrent bacteremia, more severe clinical manifestations and neurological sequelae at discharge. Based on our experience, when more abnormal CSF data, fever, and early seizure were noted in neonates with bacterial meningitis, subdural empyema should be considered, and cranial images should be arranged. 

Most of the cases with subdural empyema were treated with combined antibiotics and surgical intervention in the literature [14,15,16,19]. However, there have been cases suc-cessfully treated with a long period of antibiotics plus less invasive treatments, including aspiration, subdural washout, and drainage [20,30,38]. Another study found that 20% of children with intra-cranial abscesses received only antibiotics [39]. Although surgical interventions were performed on all our cases and final favorable outcomes were achieved, an antibiotics-only strategy should be considered in deep-seated lesions or unstable patients [40].

Some limitations in this study need to be addressed. The cohort was only from two medical centers in Taiwan, and a population-based surveillance study or data from a national database would better contribute to the generalization of the conclusions to other institutes and countries. Contrast-enhanced brain imaging is required to differentiate subdural effusion from subdural empyema, but some of our patients had only non-enhanced imaging examinations and may have been misdiagnosed with subdural empyema. Because the study period was long, the therapeutic strategies were not unified and may have been changed. Therefore, some biases are inevitable in the study. Additionally, the number of cases in this study was not adequate for performing the logistic regression analyses, and we were unable to perform the subgroup analyses as in previous studies. We suggest that further nationwide routinely collected databases related to subdural empyema after neonatal meningitis are warranted in the future.

## 5. Conclusions

In conclusion, subdural empyema is not uncommon in neonates with acute bacterial meningitis. It will cause significant neurological sequelae, although neonates with subdural empyema after acute bacterial meningitis did not have a significantly higher mortality rate in the cohort. Because most cases of subdural empyema occur several days or even weeks after the onset of meningitis, continuous monitoring of neonates with bacterial meningitis is important, which also can help the early diagnosis of other neurological complications. We found that prompt surgical intervention and prolonged use of antibiotics may help guarantee a favorable outcome. Early identification of clinical signs associated with the occurrence of long-term neurological sequelae and validation of a predictive scoring model for neonates with bacterial meningitis are warranted in the future.

## Figures and Tables

**Figure 1 antibiotics-13-00377-f001:**
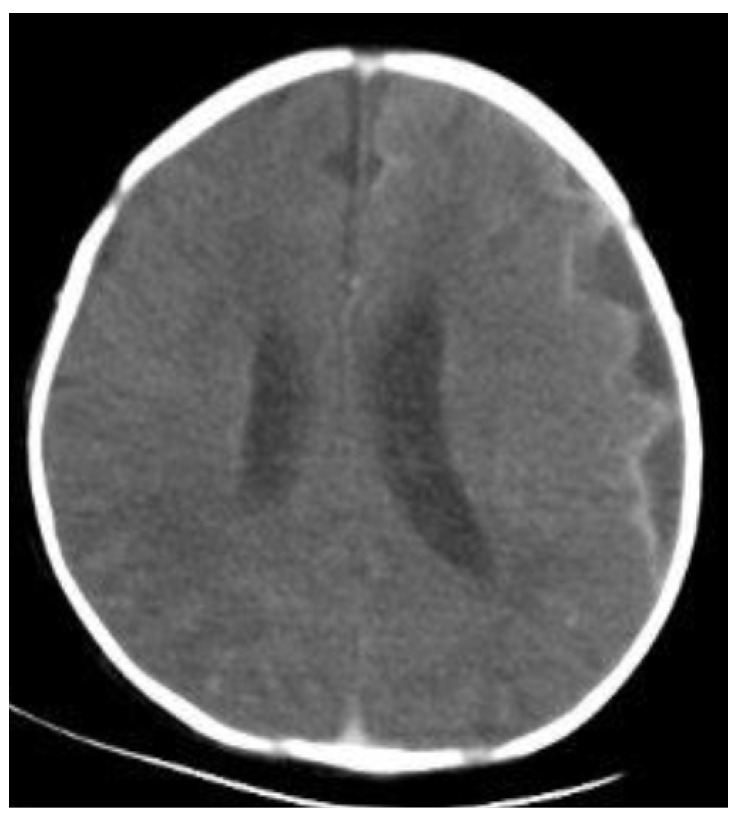
CTs of empyema complicated by bacterial meningitis in neonates. The axial CT of a neonate with bacterial meningitis with subdural empyema. This patient had a mild brain midline shift.

**Figure 2 antibiotics-13-00377-f002:**
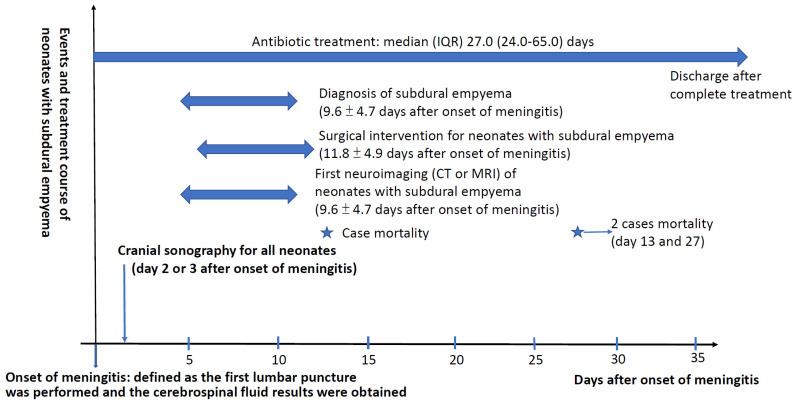
The time course of all neonates with subdural empyema, including the onset of meningitis, first neuroimaging, diagnosis of subdural empyema, and treatment courses, including surgical intervention and antibiotic treatment. The star indicates the day of the two mortality cases.

**Figure 3 antibiotics-13-00377-f003:**
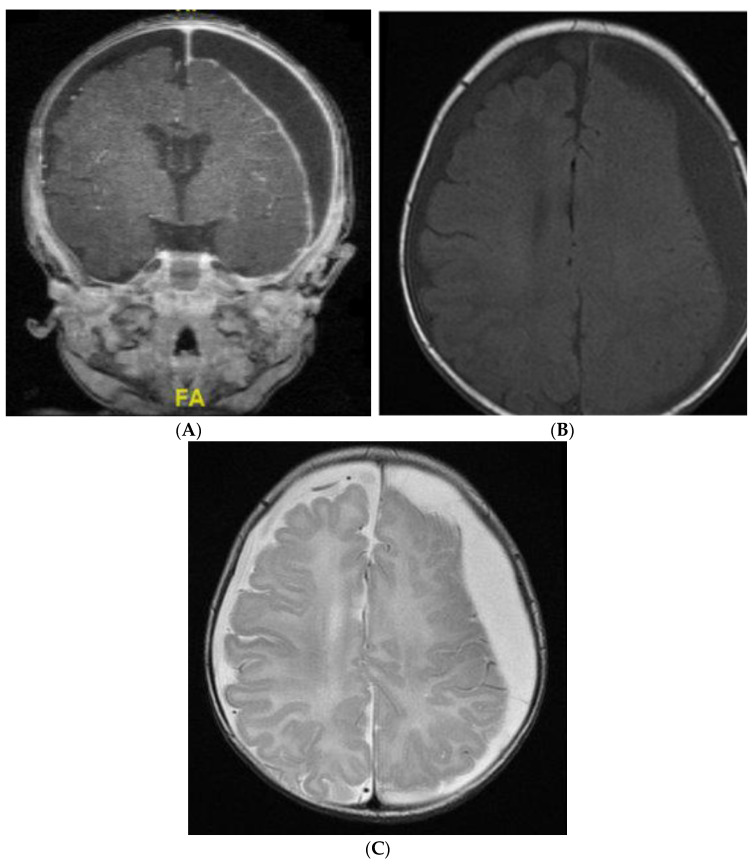
The MRI of a 3-month-old neonate with *E. coli* meningitis showed progressive bilateral subdural effusion (**A**). The apparent diffusion coefficient-weighted (**B**) and diffusion-weighted (**C**) MRIs of a subdural empyema over the left convexity are also noted.

**Table 1 antibiotics-13-00377-t001:** Patient demographics and clinical features of neonates with subdural empyema after bacterial meningitis and those without subdural empyema from Chang Gung Memorial Hospital (CGMH), 2003–2020.

	All Cases (Total *n* = 153)	Cases with Subdural Empyema after Meningitis (Total *n* = 27)	Neonates with Bacterial Meningitis (Total *n* = 126)	*p* Values
Gestational age, (week)	38.0 (36.0–39.0)	37.0 (36.0–38.0)	38.0 (36.8–40.0)	0.314
Birth body weight, (g)	2890.0 (2490–3227.5)	2870.0 (2500–3050)	2890.0 (2464.0–3240.0)	0.698
Gender, (male/female, *n*/%)	82 (53.6)/71 (46.4)	15 (55.6)/12 (44.4)	67 (53.2)/59 (46.8)	0.836
Birth by NSD/Cesarean section, *n* (%)	104 (68.0)/49 (32.0)	18 (66.7)/9 (33.3)	86 (68.2)/40 (31.7)	0.514
5 min Apgar score < 7, *n* (%)	15 (9.8)	3 (11.1)	12 (9.5)	0.628
Premature rupture of membrane, *n* (%)	27 (17.6)	5 (18.5)	22 (17.5)	0.697
Onset of bacterial meningitis (day), median (IQR)	30.0 (13.0–69.5)	30.0 (21.0–47.0)	36.0 (11.0–82.0)	0.476
Early-onset sepsis (≤7 days), *n* (%)	27 (17.6)	3 (11.1)	24 (19.0)	
Late-onset sepsis (8–90 days), *n* (%)	126 (82.4)	24 (88.9)	102 (81.0)	
Clinical features *, *n* (%)				
Fever (≥38.3 °C)	110 (71.9)	23 (85.2)	87 (69.0)	0.025
Seizure (within 3 days after onset of meningitis)	62 (40.5)	22 (81.5)	40 (31.7)	<0.001
Apnea, bradycardia and/or cyanosis	91 (59.5)	17 (63.0)	74 (58.7)	0.245
Ventilator requirement				0.131
Room air	62 (40.5)	10 (37.0)	52 (41.3)	
Nasal canula	9 (5.9)	3 (11.1)	6 (4.8)	
Non-invasive ventilator (N-CPAP and N-IMV)	16 (10.4)	4 (14.8)	12 (9.5)	
Intubation	56 (36.6)	7 (25.9)	49 (38.9)	
High-frequency oscillatory ventilator	10 (6.5)	3 (11.1)	7 (5.5)	0.053
Abdominal distension and/or vomiting	92 (60.1)	18 (66.7)	74 (58.7)	0.520
Hypoglycemia	26 (17.0)	5 (18.5)	21 (16.6)	0.782
Hypotension	54 (35.1)	12 (44.4)	42 (33.3)	0.277
Severe sepsis	73 (47.7)	16 (59.3)	57 (45.2)	0.133
Disseminated intravascular coagulopathy	31 (20.3)	7 (25.9)	24 (19.0)	0.286
Requirement of blood transfusion **	89 (58.2)	19 (70.4)	70 (55.6)	0.114
Laboratory data at onset of GBS bacteremia, *n* (%)				
Leukocytosis (WBC > 20,000/L)	79 (51.6)	18 (66.7)	61 (48.4)	0.065
Leukopenia (WBC < 4000/L)	52 (34.0)	9 (33.3)	43 (34.1)	1.000
Shift to left in WBC (immature > 20%)	35 (22.9)	8 (29.6)	27 (21.4)	0.247
Anemia (hemoglobin level < 11.5 g/dL)	80 (52.3)	16 (59.3)	64 (50.8)	0.279
Thrombocytopenia (platelet < 150,000/uL)	42 (27.5)	8 (29.6)	34 (27.0)	0.474
Metabolic acidosis	54 (35.3)	11 (40.7)	43 (34.1)	0.329
Coagulopathy	52 (34.0)	11 (40.7)	41 (32.5)	0.274
C-reactive protein (mg/dL), median (IQR)	121.0 (50.6–187.4)	116.5 (70.0–155.5)	108.0 (52.5–179.8)	0.616
Cerebrospinal fluid examinations				
WBC count (/L), median (IQR)	32.2 (11.5–480.0)	300.0 (50.0–2593.8)	25.9 (10.8–380.0)	0.011
Protein level (mg/dL), median (IQR)	274.6 (111.6–417.2)	333.0 (275.5–493.5)	233.0 (92.7–370.0)	0.006
Glucose level (mg/dL), median (IQR)	31.0 (7.0–52.0)	12.0 (5.0–30.5)	37.5 (12.8–54.3)	0.003
Pathogens				0.189
*Streptococcus agalactiae* (GBS)	54 (35.3)	12 (44.4)	42 (33.3)	
*E. coli*	39 (25.5)	9 (33.3)	30 (23.8)	
Other gram-negative bacilli	28 (18.3)	4 (14.8)	24 (19.0)	
Other gram-positive cocci	21 (13.7)	2 (7.4)	19 (15.1)	
CSF culture-negative cases	11 (7.2)	0 (0)	11 (8.7)	

All *p* values are the comparisons between neonates with subdural empyema after bacterial meningitis and those without subdural empyema; All data are expressed as numbers (%) or medians (IQR); * At the onset of bacterial meningitis; ** Including leukocyte, poor red blood cell, platelet transfusion and correlation of coagulopathy and/or disseminated intravascular coagulopathy; IQR: interquartile range; WBC: white blood cell count; N-CPAP: nasal continuous positive airway pressure; N-IMV: non-invasive mechanical ventilation.

**Table 2 antibiotics-13-00377-t002:** Neurological complications and final outcomes in neonates with subdural empyema after bacterial meningitis and those without subdural empyema in CGMH, 2003–2020.

Neurological Complications, Sequelae and Death	Neonates with Subdural Empyema after Meningitis (*n* = 27)	Neonates with Bacterial Meningitis (*n* = 126)	*p* Values
Any neurological complications	27 (100.0)	98 (77.8)	<0.001
Seizure with anti-convulsants at discharge	23 (85.2)	44 (34.9)	<0.001
Subdural effusion	22 (81.5)	42 (33.3)	<0.001
Increased intracranial pressure	8 (29.6)	32 (25.4)	0.636
Ventriculomegaly	14 (51.9)	48 (38.1)	0.280
Hydrocephalus	8 (29.6)	23 (18.3)	0.196
Encephalomalacia	4 (14.8)	11 (8.7)	0.335
Subependymal hemorrhage	2 (7.4)	21 (16.7)	0.222
Intraventricular hemorrhage	2 (7.4)	12 (9.5)	0.663
Ventriculitis	3 (11.1)	13 (10.3)	0.903
Periventricular leukomalacia	7 (25.9)	4 (3.2)	0.001
Infarction	3 (11.1)	9 (7.1)	0.486
Brain atrophy	1 (3.7)	2 (1.6)	0.187
Discharge with neurological sequelae	14 (56.6)	54 (47.8)	0.260
Final mortality *	2 (7.4)	13 (10.3)	0.463

All data are expressed as numbers (%); * Including cases of critical discharge at the family’s request.

**Table 3 antibiotics-13-00377-t003:** Risk factors for final unfavorable outcomes (death or major neurological sequelae at discharge) by univariate and multivariate analyses for neonates with bacterial meningitis.

Parameters	Univariate Analysis		Multivariate Analysis	
	OR (95% CI)	*p* Value	Adjusted OR (95% CI)	*p* Value
Preterm birth (GA < 37 weeks)	2.28 (1.08–4.81)	0.030	1.83 (0.76–4.25)	0.194
Septic shock	2.11 (1.01–4.39)	0.047	1.99 (0.58–6.84)	0.282
Respiratory failure (requirement of intubation)	2.53 (1.12–5.74)	0.026	1.35 (0.39–4.78)	0.712
Concurrent bacteremia	0.78 (0.41–1.50)	0.462		
GBS versus gram-negative bacilli	0.74 (0.38–1.67)	0.862		
High protein level in CSF	0.93 (0.46–1.87)	0.835		
Early-onset sepsis	4.26 (1.51–12.04)	0.006	3.38 (1.10–8.04)	0.027
The presence of subdural empyema	1.72 (0.83–3.55)	0.218	1.42 (0.87–2.56)	0.356
Seizure at onset	2.48 (1.28–4.81)	0.007	2.45 (1.15–4.56)	0.013
Thrombocytopenia (platelet count < 150,000/μL)	0.89 (0.41–1.91)	0.762		

GA: gestational age; CSF: cerebrospinal fluid; OR: odds ratio; 95% CI: 95% confidence interval.

## Data Availability

The datasets used/or analyzed during the current study are available from the corresponding author upon reasonable request.

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
