# Peer review of "The Epidemiology, Management and Therapeutic Outcomes of Subdural Empyema in Neonates with Acute Bacterial Meningitis"

_antibiotics, 2024, doi:10.3390/antibiotics13040377_

Round 1

Reviewer 1 Report

Comments and Suggestions for Authors

The authors provide a nice 17 year long case series of neonatal subdural empyema from two hospitals in Taiwan

Abstract:

- “timely surgical intervention to optimize the outcomes are urgently needed” => this sentence is unclear. Do the authors advise early surgical intervention, or do they mean that a future study priority should be to determine when/for whom surgical intervention improves outomec?

Introduction:

- minor: “Ne[1]onates with bacterial meningitis are associated” => meningitis is associated with, but neonates have high riks

- minor: I do not think that empyema has the capability of threatening clinicians

Methods:

- minor: “Neonates with central nervous system (CNS) infections after artificial devices, including ventriculoperitoneal shunt, extraventricular drainage and post-neurosurgical CNS infections were not enrolled in this study” => I think the authors mean “excluded” instead of “not included”

- minor: “X-ray doctor” => radiologist?

- “The trend of proportions of the categorical variables among the subgroups was analyzed by the Cochran-Armitage trend test” => can the authors please argue why they want to use this test and for what analysis?

- “Covariates presumed to be associated with final adverse outcomes based on previous studies and those associated with mortality at P< 0.1 were subsequently entered into the multivariable logistic regression model.”=> what co-variates were a priori chosen? How many? Based on which studies?

Results:

- “During the study period, a total of 153 neonates with bacterial meningitis were identified from the NICUs of the two medical centers in Taiwan” => what is the resulting (order of) the estimate of bacterial meningitis incidence (rate per year per 1000 nicu patients), and what does this suggest about the proportion of bacterial meningitis cases that were retrospectively identified?

- Please add to the first/second paragraph How many of the bacterial meningitis cases had neuroimaging performed and the median (iqr) of the number of days between the lumbar puncture and first neuro-imaging.

- please provide a brakedown of all identified pathogens for the empyema and non-empyema groups.

- how many patients were csf culture positive? How many had positibe blood cultures? And was a positive csf culture associated with development/presence of empyema?

- When were the lp results obtained in relation to the date the subdural effusion was identified? Do the authors think that “higher white blood cell counts, lower glucose levels and higher protein levels” reflect the presence of empyema, of are prognostic for the subsequent development of empyema?

- Table 1. Do the authors mean to show the  white cell count per litre or microlitre? (WBC count (/L))

 - “There were 2 patients (7.4%) died and the mortality rate was not significantly higher.” => p value?

- “Neonates with early-onset disease, seizure at onset (defined as seizure attack within 48 hours after onset of meningitis), respiratory failure and septic shock, the presence of subdural empyema that required surgical intervention were enrolled into the multivariable regression model.” => why was only a subselection included in this analysis?

- the time course of the epyema cases could be presented in more detail. Possibly a figure with a time-line for each patient that includes start of bacterial meningitis, LP, start and end of antibiotic treatment, day first neuro-imaging, day of empyema diagnosis, surgical intervention, discharge or death

- how many patients had neurosurgical interventions? How did these patients differ form empyema patients that did not have neurosurgical operations?

Discussion:

- “We found that prompt surgical intervention and prolonged use of anti[1]biotics may help to guarant” => what is the basis in the results section for this conclusion?

Comments on the Quality of English Language

needs work

Author Response

Dear reviewer:

      Please see the attachment. I appreciate your review and comments, thank you.

Best regard,

Tsai Ming Horng

Abstract:

- “timely surgical intervention to optimize the outcomes are urgently needed” => this sentence is unclear. Do the authors advise early surgical intervention, or do they mean that a future study priority should be to determine when/for whom surgical intervention improves outcomes?

Reply:

       Thank you for your constructive advice. Because most of the patients with subdural empyema were treated with combined antibiotics and surgical intervention and we had comparable outcomes, timely surgical intervention to optimize the outcomes are suggested. To make it more clear, I will revise the sentence to be “appropriate long-term antibiotic treatment after surgical intervention are required to treat the infectious process.” (the last sentence of the abstract), thank you.

Introduction:

- minor: “Neonates with bacterial meningitis are associated” => meningitis is associated with, but neonates have high riks

Reply:

       Thank you for your constructive advice. I will revise the sentence to be “Bacterial meningitis is associated with a high risk of neurological complications and neonates with meningitis are at high risk of long term neurological sequelae”, (the 2nd line of the first paragraph of the introduction in page 2), thank you.

- minor: I do not think that empyema has the capability of threatening clinicians

Reply:

       Thank you for your constructive advice. I will revise the sentence to be “a prolonged course of treatment and hospitalization”, (the 6th line of  the 2nd paragraph of the introduction section, in page 2), thank you.

Methods:

- minor: “Neonates with central nervous system (CNS) infections after artificial devices, including ventriculoperitoneal shunt, extraventricular drainage and post-neurosurgical CNS infections were not enrolled in this study” => I think the authors mean “excluded” instead of “not included”

Reply:

Thank you for your constructive advice. I will revise “not included” to “excluded” in the revised manuscript, thank you. (the 13rd line of the 1st paragraph in the method section, page 2)

- minor: “X-ray doctor” => radiologist?

Reply:

Thank you for your constructive advice. I will revise the “x-ray doctor” to be “radiologist”, thank you. (the first line of the 2nd paragraph in the method section, page 2)

- “The trend of proportions of the categorical variables among the subgroups was analyzed by the Cochran-Armitage trend test” => can the authors please argue why they want to use this test and for what analysis?

Reply:

Thank you for your constructive advice. Actually we finally did not use the test in the study. We tried to use this in the beginning, but we did not do it because this cannot give us more results. I will delete the sentence in the revised version of the manuscript, thank you.

- “Covariates presumed to be associated with final adverse outcomes based on previous studies and those associated with mortality at P< 0.1 were subsequently entered into the multivariable logistic regression model.”=> what co-variates were a priori chosen? How many? Based on which studies?

Reply:

Thank you for your questions. This question is similar to that mentioned in the result section. Basically we would enroll covariates based on our previous studies on ref. no. 3, 12, 21 and 34. I will cite the references in the revised manuscript, thank you. (in page 4, the 6th line of the first paragraph)

Results:

- “During the study period, a total of 153 neonates with bacterial meningitis were identified from the NICUs of the two medical centers in Taiwan” => what is the resulting (order of) the estimate of bacterial meningitis incidence (rate per year per 1000 NICU patients), and what does this suggest about the proportion of bacterial meningitis cases that were retrospectively identified?

Reply:

Thank you for your constructive advice. Because the data were from two medical centers for an extensive period of time (18 years), we cannot catch exactly the overall patient numbers and cannot provide the data of bacterial meningitis incidence (rate per year per 1000 NICU patients). In my opinion, the meningitis incidence is very low.

The proportion of bacterial meningitis cases in the NICU was not very high. Although uncommon, bacterial meningitis in the NICU is associated with high mortality and morbidity rates.

- Please add to the first/second paragraph How many of the bacterial meningitis cases had neuroimaging performed and the median (iqr) of the number of days between the lumbar puncture and first neuro-imaging.

Reply:

Thank you for your constructive advice. I will add the above information in the last sentence of 2nd paragraph of the result sections, in page 4, as the following: All neonates with meningitis had a cranial sonography examination following bacterial meningitis, and cranial imaging studies were performed in 98 (64.1%) of 153 patients, and the median (IQR) days between the lumbar puncture and first neuroimaging was 12.5 (7.0-18.0) days. thank you.

- please provide a brakedown of all identified pathogens for the empyema and non-empyema groups.

Reply:

Thank you for your instructive advice. I will provide a brakedown of all identified pathogens for the empyema and non-empyema groups in the button of table 1 (page 6), thank you. Because the table 1 is too long, I just summarize them as GBS, E coli, other Gram-negative bacilli and other gram-positive cocci, thank you.

- how many patients were csf culture positive? How many had positive blood cultures? And was a positive csf culture associated with development/presence of empyema?

Reply:

Thank you for your questions. In overall cohort, there were a total of 11 patients who had negative CSF culture, so 92.8% (n=142) patients had positive CSF culture. In overall cohort, 128 (83.7%) neonates had concurrent bacteremia (positive blood culture). All the patients with subdural empyema had positive CSF culture, but we cannot conclude that a positive CSF culture to be associated with development of empyema, because most of the patients without empyema also had positive CSF culture.

 I will add the above information in the revised manuscript as the following: In the overall cohort of neonates with meningitis, 92.8% (n=142) patients had a positive CSF culture and 128 (83.7%) neonates had concurrent bacteremia (positive blood culture). All neonates with subdural empyema had positive CSF cultures. (the 1st line of the 2nd paragraph of the result section, page 4.

- When were the lp results obtained in relation to the date the subdural effusion was identified? Do the authors think that “higher white blood cell counts, lower glucose levels and higher protein levels” reflect the presence of empyema, of are prognostic for the subsequent development of empyema?

Reply:

Thank you for your questions. The Lumbar puncture results were most obtained nearly the onset of meningitis, without the first day after onset of meningitis. Actually we usually defined onset of meningitis as the day of lumbar puncture performed. In the result section, I have mentioned that “The median age at onset and diagnosis of bacterial meningitis and subdural empyema was 30 (21.0-47.0) [Interquartile range (IQR)] and 38 (25.0-60.0) days old, respectively. The average occurrence of subdural empyema from onset of acute meningitis was 9.6 ± 4.7 days.” (the middle of the first paragraph of the result section, page 4. I will not especially mention when were the lp results obtained in relation to the date subdural empyema was identified.

Based on this study, we can conclude that the CSF results of neonates with subdural empyema showed a significantly higher white blood cell counts, lower glucose levels and higher protein levels (p = 0.011, 0.003 and 0.006, respectively). However, we will not conclude these parameters to reflect the presence of empyema or prognostic for the subsequent development of empyema because we did not have enough cases to confirm the findings. I have mentioned this issue in the discussion section, page 9, the first paragraph, thank you.

- Table 1. Do the authors mean to show the  white cell count per liter or microliter? (WBC count (/L))

Reply:

Thank you for your question. The data in Table 1 is the white cell count per liter (WBC count/L), thank you.

 - “There were 2 patients (7.4%) died and the mortality rate was not significantly higher.” => p value?

Reply:

Thank you for your question. The p value is 0.463 (Table 2) when compared with those with meningitis without subdural empyema, thank you.

- “Neonates with early-onset disease, seizure at onset (defined as seizure attack within 48 hours after onset of meningitis), respiratory failure and septic shock, the presence of subdural empyema that required surgical intervention were enrolled into the multivariable regression model.” => why was only a subselection included in this analysis?

Reply:

Thank you for your questions. Because we have had the initial evaluation, and applied the previous studies that have investigated the risk factors associated with final adverse outcomes. I cite the references in the statistical analyses section of the revised manuscript, thank you. (in page 4, the 6th line of the first paragraph)

- the time course of the empyema cases could be presented in more detail. Possibly a figure with a time-line for each patient that includes start of bacterial meningitis, LP, start and end of antibiotic treatment, day first neuro-imaging, day of empyema diagnosis, surgical intervention, discharge or death

Reply:

Thank you for your constructive advice. I will add a figure to list the time-line for the patients with subdural empyema and includes all these events. I will add The time course of all neonates with subdural empyema was summarized in Figure 2. (the result section page 7, the first paragraph), thank you.

- how many patients had neurosurgical interventions? How did these patients differ from empyema patients that did not have neurosurgical operations?

Reply:

Thank you for your constructive advice. All the patients with subdural empyema received surgical intervention. In neonates with meningitis, 39 patients (25.4%) received surgical treatment. In neonates without subdural empyema, surgical treatments were mainly VP-shunt or extraventricular drainage for hydrocephalus. Because there was no patient with subdural empyema that did not have neurosurgical operations, we cannot find the patients differ from them.

Because this study focused on neonates with subdural empyema, I will not mention this issue in the revised manuscript, thank you.

Discussion:

- “We found that prompt surgical intervention and prolonged use of antibiotics may help to guarantee” => what is the basis in the results section for this conclusion?

Reply:

Thank you for your constructive advice. Because all our patients with subdural empyema received surgical intervention and a longer duration of antibiotics than the controls were required, we had the conclusion. Because neonates with subdural empyema had an acceptable final outcome when compared with the controls, we had the above conclusion, thank you.

Reviewer 2 Report

Comments and Suggestions for Authors

Dear authors,

 I have read with interest your manuscript titled " The epidemiology, management and therapeutic outcomes of subdural empyema in neonates with acute bacterial meningitis." I believe that this is a well-constructed manuscript with only a few minor issues that need to be addressed.

1.    Figure 1 It is mentioned that it is the CTs of two neonates and only the image of 1 appears.

 2.    Please explain the rationale behind selecting all instances in the control group rather than randomly choosing two or three times the number of cases within that group.

 3.    It would be interesting to include a table of resistance patterns of microbiological isolates.

Author Response

Dear reviewer

      Please see the attachment. I appreciate your review and comments, thank you.

Best regard,

Tsai Ming Horng
Dear authors,

 I have read with interest your manuscript titled " The epidemiology, management and therapeutic outcomes of subdural empyema in neonates with acute bacterial meningitis." I believe that this is a well-constructed manuscript with only a few minor issues that need to be addressed.

  1. Figure 1 It is mentioned that it is the CTs of two neonates and only the image of 1 appears.

Reply:

Thank you for your instructive advice. There is only one neonate in figure 1. I am sorry for the mistake, and we will correct the mistake in the revised manuscript, thank you.

  1. Please explain the rationale behind selecting all instances in the control group rather than randomly choosing two or three times the number of cases within that group.

Reply:

 Thank you for your questions. We selected all instances in the control group rather than randomly choosing two or three times the number of cases because we wanted to avoid selection bias. Additionally, there were not so many cases to have meningitis in the study period, so we select all the instances in the controls.

  1. It would be interesting to include a table of resistance patterns of microbiological isolates.

Reply:

       Thank you for your instructive advice. Because most of the isolates were GBS and E coli, and all the GBS isolates were sensitive to all antibiotics (ampicillin….), it may not be necessary to list a table of resistance patterns of microbiological isolates.

Round 2

Reviewer 1 Report

Comments and Suggestions for Authors The two most relevant points were 1. the authors conclude that all neonates with empyema require neurosurgery. Although as a medical doctor, I agree with the authors that this is probably the best treatment for most (and possibly all) of these patients, this conclusion cannot be based on the results from this study. Just because ''X" was done in all cases and "Y" followed is no proof. Especially because there are cases of pediatric subdural empyema that have successfully been treated with antibiotics alone. I would suggest to stick to describing the treatment and outcome and in the discussion refer to other studies
2. in the multivariable regression analysis it seems that the authors first make a subselection of the whole cohort. I do not understand why and do not think that this is the correct analysis. All cases with meningitis should be included to find prognostic predictors of who has a higher risk to develop empyema. Possibly the authors try to make a different point here, for instance about what variables/predictors were selected. However, this is unclear from the current text. It reads as if a subselection of patients is selected.

Comments on the Quality of English Language

needs work

Author Response

Dear reviewer:

     I appreciate your review and comments, thank you.

Best regard,

Tsai Ming Horng

The two most relevant points were

  1. the authors conclude that all neonates with empyema require neurosurgery. Although as a medical doctor, I agree with the authors that this is probably the best treatment for most (and possibly all) of these patients, this conclusion cannot be based on the results from this study. Just because ''X" was done in all cases and "Y" followed is no proof. Especially because there are cases of pediatric subdural empyema that have successfully been treated with antibiotics alone. I would suggest to stick to describing the treatment and outcome and in the discussion refer to other studies

Reply:

Thank you for your constructive advice. I avoid to conclude that all neonates with empyema require neurosurgery. Please see the conclusion in the abstract: Closely monitor the occurrence of subdural empyema is required and appropriate long-term antibiotic treatment after surgical intervention would lead to optimized outcomes.

I will also add a paragraph in the discussion about the treatment and outcome of subdural empyema as the following: Most of the cases with subdural empyema were treated with combined antibiotics and surgical intervention in the literature [14-16,19]. However, there have been cases successfully treated with a long period of antibiotics plus less invasive treatments, including aspiration, subdural washout, and drainage [30,36,40]. Another study found that 20% children with intra-cranial abscess received only antibiotics [41]. Although surgical interventions were performed on all our cases and final favorable outcomes were achieved, antibiotics-only strategy should be considered in deep-seated lesions or unstable patients [42]. (page 9, the last paragraph~ page 10)

  1. In the multivariable regression analysis it seems that the authors first make a subselection of the whole cohort. I do not understand why and do not think that this is the correct analysis. All cases with meningitis should be included to find prognostic predictors of who has a higher risk to develop empyema. Possibly the authors try to make a different point here, for instance about what variables/predictors were selected. However, this is unclear from the current text. It reads as if a subselection of patients is selected.

Reply:

Thank you for your constructive advice. Actually we used the whole cohort in the multivariable regression analysis.

Please notice the title of Table 3: Risk factors for final unfavorable outcomes (death or major neurological sequelae at discharge) by univariate and multivariate analysis for neonates with bacterial meningitis. I mark the title here. We aimed to investigate whether presence of subdural empyema would affect the outcomes, so all neonates with meningitis were included in the analysis.

I do not find the text where I led to misunderstanding that a subselection of patients is used in the analysis.
